# Global ground strike point characteristics in negative downward lightning flashes — part 1: Observations

Dieter R. Poelman[1], Wolfgang Schulz[2], Stephane Pedeboy[3], Dustin Hill[4], Marcelo Saba[5], Hugh Hunt[6], Lukas Schwalt[7], Christian Vergeiner[7], Carlos T. Mata[4], Carina Schumann[6], and Tom Warner[8]

[1]Royal Meteorological Institute of Belgium, Brussels, Belgium

[2]Austrian Lightning Detection and Information System (ALDIS), Vienna, Austria

[3]Météorage, Pau, France

[4]Scientific Lightning Solutions LLC (SLS), Titusville, Florida, USA

[5]National Institute for Space Research, INPE, São José dos Campos, Brazil

[6]The Johannesburg Lightning Research Laboratory, School of Electrical and Information Engineering, University of Witwatersrand Johannesburg, Johannesburg, South Africa

[7]Institute of High Voltage Engineering and System Performance, Graz University of Technology, Graz, Austria

[8]ZT Research, Rapid City, South Dakota, USA

*Correspondence to*: Dieter R. Poelman (dieter.poelman@meteo.be)

**Abstract** Information about lightning properties are important in order to advance the current understanding of lightning, whereby the characteristics of ground strike points (GSPs) are in particular helpful to improve the risk estimation for lightning protection. Lightning properties of a total of 1174 negative downward lightning flashes are analyzed. The high-speed video recordings are taken in different regions, including Austria, Brazil, South Africa and U.S.A., and are analyzed in terms of flash multiplicity, duration, interstroke intervals and ground strike point properties. According to our knowledge this is the first simultaneous analysis of GSP properties in different regions of the world applying a common methodology. Although the results vary among the data sets, the analysis reveals that a third of the flashes are single-stroke events, while the overall mean number of strokes per flash equals 3.67. From the video imagery an average of 1.56 GSPs per flash is derived, with about 60% of the multiple stroke flashes striking ground in more than one place. It follows that a ground contact point is struck 2.35 times on average. Multiple-stroke flashes last on average 371 ms, whereas the geometric mean (GM) interstroke interval value preceding strokes producing a new GSP is about 18% greater than the GM value preceding subsequent strokes following a pre-existing lightning channel. In addition, a positive correlation between the duration and multiplicity of the flash is presented. The characteristics of the subset of flashes exhibiting multiple GSPs is further examined. It follows that strokes with stroke order of two create a new GSP in 60% of the cases, while this percentage quickly drops for higher order strokes. Further, the possibility to form a new lightning channel to ground in terms of the number of strokes that conditioned the previous lightning channel shows that approximately 88% developed after the occurrence of only one stroke. Investigating the time intervals in

the other 12% of the cases when two or more strokes re-used the previous lightning channel showed that the average interstroke time interval preceding a new lightning channel is found to be more than twice the time difference between strokes that follow the previous lightning channel.

## 1 Introduction

Cumulonimbus clouds are the birthplace of one of Earths' true spectacles in nature: the lightning discharge. The development of these clouds involves a number of steps. As the building phase comes to an end, characterized by a rapid increase of growth of the clouds' height through the rise of pockets of warm and moist air, it sets the stage for super cooled cloud droplets to coagulate and increase in both mass and size. The subsequent mature phase provides the electric charge structure through a range of collisions between the icy particles. Typically, this results in the top of the cloud being predominantly positively

charged, while the bottom of the cloud accomodates the bulk of the negatively charged particles. It is at this magical moment, when eventually the difference in charge potential reaches a certain threshold, that the cloud 'switches on the light' and powerful electrical discharges appear, proudly drawing the attention of the spectator to an even greater extent than was the case moments before. Followed by the dissipation phase, this gigantic wasteland of energy, once capable of producing severe weather at ground, disappears and leaves us in awe.

Lightning radiates its energy in almost the full range of the electromagnetic spectrum. Hence, to observe and further increase our understanding of lightning discharges in these cauliflower-like clouds, and the associated forces and physical processes that are present within them, a whole range of instruments and techniques are at our disposal. The use of ground-based lightning location systems (LLS), much in the same way compared to those constructed by todays' standards, was first introduced more than 40 years ago (Lewis et al., 1960; Krider et al., 1976). Present-day LLSs operate from very low frequencies (VLF) and to

very high frequencies (VHF) and are able to detect cloud-to-ground (CG) strokes and intracloud (IC) pulses (e.g., Bürgesser, 2017; Said et al., 2010; Gaffard et al, 2008; Zhu et al., 2017; Murphy et al., 2021; Schulz et al., 2016; Coquillat et al., 2019). Depending on the adopted technique, the total pathway covered by a lightning flash can be presented as a single point or constitute several (even up to thousands of points) for a single discharge. Modern ground-based low frequency LLSs are capable of differentiating between CG and IC flashes and tend to perform well in terms of flash and stroke detection

efficiencies, providing the location of downward CG ground strike points with high confidence.

On the other hand, satellite missions with onboard dedicated instruments provide a different way of capturing the stroboscopical dance of lightning discharges by observing the scattered light peaking through the top of the cloud. The signature of the strong optical oxygen triplet emission line at 777.4 nm is typically what is observed by means of specifically designed cameras. Although first attempts started already from the 1970s (Vorpahl J.A. et al. 1970; Sparrow & Ney, 1971;

Turman, 1978), one had to wait until 1995 with the launch of the OV-1 (MicroLab 1) satellite with the onboard Optical Transient Detector (OTD), closely followed by the Tropical Rainfall Measuring Mission (TRMM) carrying the Lightning Imaging Sensor (LIS) in 1997, to witness the potential and significance of those type of missions. While the latter satellites

moved in a polar orbit around the Earth, the latest and future type of optical lightning instruments are being put in operation from geostationairy orbit (Goodman et al., 2013; Yang et al., 2017, Grandell et al., 2009), therewith expanding even further

the range of associated applications.

Even though its not uncommon to become lyrical about todays' achievements in this field of research, the observations from ground-based LLSs as well as from space have, besides governing many advantages, one fundamental drawback as the lightning discharges are observed indirectly. By contrast, high-speed camera observations observe the light emitted directly by the lightning discharge, thereby documenting the flow of the electrical charged particles through the air and provide, linked

to electric field measurements, a means to investigate in great detail the associated optical and electromagnetic properties of natural downward lightning flashes. With frame rates of 200 per second (fps) or more, the different strokes that compose a multi-stroke flash can each be captured individually, while it is the electric field measurement that undisputably identifies the polarity of each stroke. Furthermore, video imagery enable us to determine, if not too distant and/or obscured by precipitation, whether each individual stroke creates a new ground contact point (NGC) or follows a pre-existing lightning channel (PEC).

The characteristics deduced from this is not only relevant from a pure scientific perspective, but is essential in developing adequate lightning protection solutions as the level of lightning protection and risk to be mitigated is derived from the density of lightning terminations in a region. Typically, this is based on flash density values but there have been recommendations to increase calculated densities by a factor of two to account for multiple ground strike point flashes (Bouquegneau, 2013, 2014; IEC 62858 Ed. 2, 2019). Understanding these characteristics is essential for evaluating whether such a factor is relevant.

In this paper, high-speed camera observations are analyzed in order to deduce some of the characteristics observed in natural negative downward lightning flashes. Section 2 describes briefly the instrumentation used per region and analysis thereof is provided in Section 3. Section 4 summarizes the findings of this study. In this context, it is worthwhile mentioning that the data sets described here serve as basis to investigate the ability of so-called ground strike point algorithms to correctly group strokes in flashes according to the observed ground strike points (Poelman et al., 2021, nhess-2021-13 companion paper).

**2 Data Acquisition and Analysis**

Ground-truth campaigns are time consuming in order to gather enough data to be statistically relevant. To reach this objective, ground-truth datasets are collected from different geographical regions and taken over various periods in time: Austria (AT) in 2012, 2015, 2017 and 2018, Brazil (BR) in 2008, South Africa (SA) in 2017-2019 and U.S.A. (US) in 2015.

Before going into more detail on the methods of data collection, it is of importance to recognize the limitations inherent to

high-speed camera observations when used in flash characteristic studies. In particular, strokes creating a new termination could be missed by the camera when occurring out of the camera's field of view. In addition, the record length should be long enough in order to capture the entire flash, i.e., typically longer than one second. Aiming to minimize as much as possible the influence of the latter on the retrieved flash statistics, high-speed camera observations should be checked against concurrent electric field measurements to ensure a stroke was not missed. In this, flashes with lightning channels that are outside the field

of view should be excluded from the data. For the measurements in all of the data sets presented in this study electric field measurements have been used, and therefore only flashes where a clear visible lightning channel to the ground is observed for all the associated strokes are included. However, it should be noted that even though such a selection of flashes is made, it does not undeniably resolve the true contact point all of the time. This is certainly true when the observations are made at ground level. As such, the amount of ground strike points retrieved from the video measurements as discussed later on in this

study should be regarded as a lower limit.

Finally, it is essential to remark that the flash grouping, i.e., grouping strokes belonging to the same flash, is based on the video images alone without any input from LLS data whatsoever. Clearly, it would make more sense to trace the lightning leader back to the location of the preliminary breakdown and only group strokes that emanate from a common charge region. However, this would require observations made by a Lightning Mapping Array.

In what follows, a description is given of the instrumentation set-up used at the different regions and the periods of investigation.

### 2.1 Austria

A so-called video and field recording system (VFRS) is used to document lightning strikes in the alpine region of Austria. The VFRS consists of a high-speed camera and an electric field measurement system and both are GPS time synchronized. The

system is composed of a flat plate antenna, an integrator and an amplifier, a fiber optic link, a digitizer and a PXI system (Schulz et al., 2005). The camera used for the data recorded in 2015, 2017 and 2018 is the Vision Research Phantom V9.1, operated at a frame rate of 2000 frames per second (fps), a 14-bit image depth and a resolution of $1248 \times 400$ pixels (Schulz et al., 2009; Vergeiner et al., 2016, Schwalt, 2019, Schwalt et al., 2020) with a total record length of 1.6 s. In 2012 a monochrome (8 bit per pixel) Basler camera was used at 200 fps with VGA resolution, i.e., 640 x 480 pixels, and with a record

length of 5 s.

### 2.2 Brazil

A Photron PCI-512 high-speed digital camera, operating at 4000 fps, was used to record the flashes in Southeastern Brazil in 2008. The high-speed video images are GPS time-stamped to an accuracy better than 1 millisecond with a 1 s pre-trigger time and a total recording time of 2 s. Each trigger pulse was initiated manually by an operator when a flash was observed within

**Table 1.** Flash characteristics

| Parameter | Location ground-truth observations | | | | |
|---|---|---|---|---|---|
| | AT | BR | SA | US | ALL |
| $N$(flashes) | 490 | 122 | 484 | 78 | 1174 |
| $N$(strokes) | 1539 | 619 | 1839 | 305 | 4302 |
| Mean multiplicity | 3.14 | 5.07 | 3.8 | 3.90 | 3.67 |
| Max multiplicity | 14 | 17 | 26 | 14 | 26 |
| Percentage of single stroke flashes | 29.2 | 23.0 | 38.4 | 25.6 | 32.1 |
| $N$(GSP) | 845 | 232 | 626 | 129 | 1832 |
| Average $N$(GSP/flash) | 1.72 | 1.90 | 1.29 | 1.65 | 1.56 |
| Max $N$(GSP/flash) | 5 | 4 | 5 | 4 | 5 |
| Average $N$(strokes/GSP) | 1.82 | 2.67 | 2.94 | 2.36 | 2.35 |
| Average flash duration[1,2] (ms) | | | | | |
| All flashes | 233 | 415 | 262 | 236 | 264 |
| Multiple-stroke flashes | 306 | 538 | 394 | 328 | 371 |
| Occurrence of forked strokes[3] | | | | | |
| Percentage of flashes at least 1 forked stroke | 9.4 | 10.7 | 7.0 | 10.3 | 8.3 |
| Percentage of forked strokes in flashes containing at least 1 forked stroke | 34.4 | 21.8 | 20.8 | 42.8 | 24.1 |
| Percentage of forked strokes in the overall data set | 3.7 | 2.3 | 2.2 | 2.9 | 2.5 |
| Continuing Current (CC) | | | | | |
| Mean (ms) | 67.1 | 36.5 | 38.5 | / | 44.5 |
| Median (ms) | 15.0 | 8.0 | 9.0 | / | 10.0 |
| Max (ms) | 540 | 705 | 929 | / | 929 |
| Percentage of strokes followed by CC $\geq$ 3 ms | 33.7 | 71.7 | 73.0 | / | 57.7 |
| Percentage of strokes followed by CC $\geq$ 500 ms | 0.26 | 0.32 | 0.38 | / | 0.33 |
| Percentage of flashes containing CC $\geq$ 10 ms | 37.8 | 61.5 | 61.8 | / | 51.0 |

[1] Flash duration is defined as the time interval between the occurrence of the first return stroke and the end of the continuing current following the last return stroke, if present.

[2] Values for US do not include continuing current duration.

[3] For AT only based on data taken in 2018.


the camera field-of-view. For more details on the operation and accuracy of high-speed cameras for lightning observations, see Saba et al. (2016). The polarity of the strokes is determined by matching the strokes to electric field measurements and to the observations of a local lightning location system BrasilDat in Brazil. More information on the characteristics of this network is given by Naccarato and Pinto (2009).

## 2.3 South Africa

The high-speed study of lightning flashes over Johannesburg, South Africa began in 2017. Johannesburg is located in the North Eastern province of Gauteng and sits at an altitude of approximately 1600 m above sea level (asl). The area has seasonal thunderstorms, generally occurring during the mid-to-late afternoons in the summer months (September-April, Southern Hemisphere) and with no thunderstorm activity during the winter months. The area has a flash density of 15 to 18 flashes/km$^2$/year (Evert, 2017). The setup utilizes two high-speed cameras (a Phantom v7.1 and a Phantom v310) which are located North-West of the city. Frame rates used are in the range of 5000 to 15000 fps and all captured videos are GPS time-stamped. A 1.8 second buffer time is used and events are manually triggered. Typically, the pre-trigger and post-trigger were set approximately 60/40 of the 1.8 second buffer, respectively. Note that in this area both downward and upward lightning discharges are captured. The latter are events triggered by the two tall towers located in Johannesburg – the Sentech and Hillbrow tower, approximately 250 m each  (Schumann, 2018). However,  all tower events in the SA data set are excluded from the analysis in this study.

## 2.4 U.S.A.

The observations used in this study are taken from the Kennedy Space Center/Cape Canaveral Air Force Station (KSC/CCAFS) in 2015 (Hill et al., 2016). A compact network of 13 high-speed cameras record cloud-to-ground lightning return strokes terminating on KSC/CCAFS property, with geographic emphasis on the areas surrounding Launch Complex 39B (LC-39B), Launch Complex 39A (LC-39A), Launch Complex 41 (LC-41), and the Vehicle Assembly Building (VAB). Eight of the cameras are located on tall structures at altitudes greater than 150 m, providing downward vantage points. Many of the cameras are configured with intersecting fields of view to provide multi-angle scenes of the same discharge. The high-speed cameras sample at either 3,200 or 16,000 fps. The cameras have memory segment lengths ranging from about 100 ms to 400 ms and operate in segmented memory mode in order to capture many consecutive events without overrunning the internal buffer. In this way, the entire sequence of strokes is captured over the full duration of a flash. In addition, six wideband rate of change of electric field (dE/dt) sensors provide information on the polarity of the discharges. The digitization time bases of these

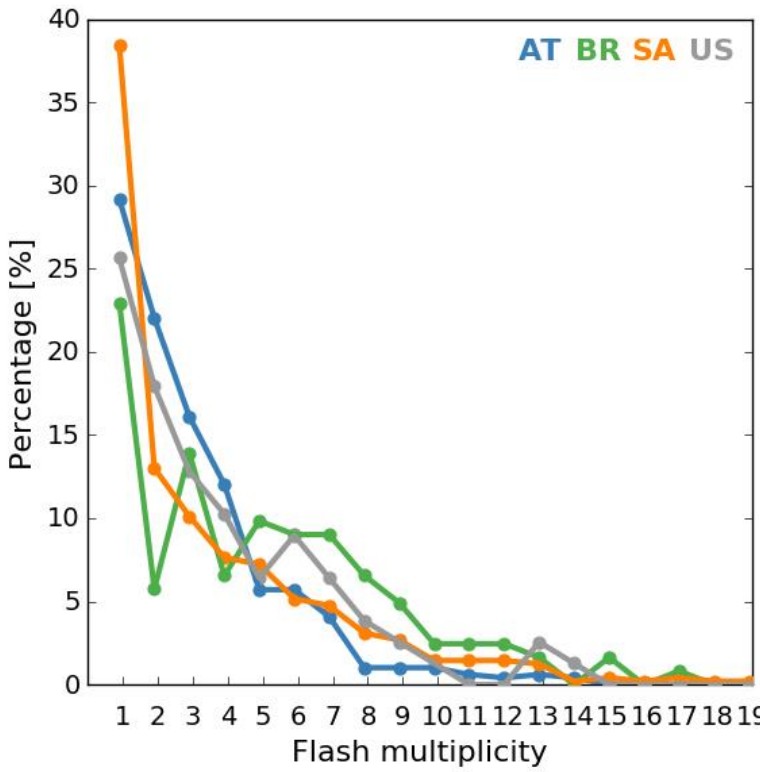

**Figure 1: Distribution of the number of strokes per flash.**

geographically independent sensors are synchronized with RMS accuracy of 15 ns.

## 3. Results

The combined data sets comprise of 1174 flashes and 4302 strokes. The characteristics of each individual data set regarding flashes, strokes, ground strike points, forked stroke occurrence, multplicity, flash duration and length of the continuing current (CC) are presented in Table 1. The largest data set in terms of amount of flashes is the one of Austria with 490 flashes, closely followed by the South African data set containing 484 flashes. On the other hand, the data set of South Africa includes by far the largest amount of strokes. The distribution of the flash multiplicity of the individual data sets is depicted in Figure 1. Clearly, the flash multiplicity depends on the ability to identify all the respective strokes that occurred during the flash. The video frame rates listed in the previous section that were used for the observations are believed to be more than sufficient to meet this requirement. Mean flash multiplicities range from 3.14 (AT) to 5.07 (BR) strokes per flash, with an observed overall combined flash multiplicity of 3.67. The multiplicities in this study are in line with average multiplicity values published in other studies such as Rakov et al. (1994), Cooray and Perez (1994), Cooray and Jayaratne (1994), Saba et al. (2006), Saraiva et al. (2010) and lower than what was found by Ballarotti et al. (2012) and Kitagawa et al. (1962). From Fig. 1 and Table 1 it

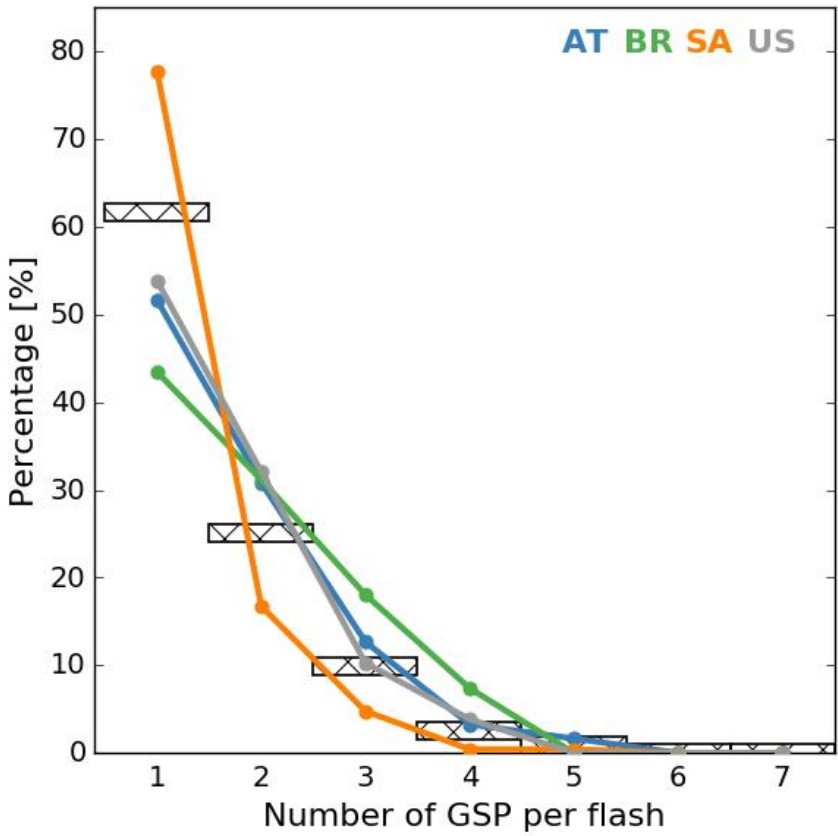

Figure 2: Distribution of the number of GSPs per flash. The shaded rectangles represent the result for the combined data sets.

can be seen that the percentage of single stroke flashes varies in between 23% (BR) and 38.4% (SA), with an average of 32.1% for all the flashes combined. One could argue that the latter percentages are somewhat higher compared to those quoted in well-known reports of accurate stroke count studies such as the 13% observed in New Mexico by Kitagawa et al. (1962), 17% in Florida by Rakov and Uman (1990), 18% in Upssala by Cooray and Pérez (1994), and the 21% in Sri Lanka as described in Cooray and Jayaratne (1994). Nonetheless, the 29.2% retrieved for AT in this study is comparable to the 27% analyzed in detail by Schwalt et al. (2021), which, in addition, also demonstrated that the percentage of single stroke flashes can vary considerably from one storm to the other without an apparent dependency on thunderstorm type or underlying meteorological characteristics. The 23% of single-stroke flashes for BR in the present study is only a few percent higher than the 17% observed within the Sao Paolo State retrieved by Ballarotti et al. (2012). In case of SA, there exist no previously published values of single stroke occurrences against which to check the 38.4%. It seems that this area, at an altitude of about 1600 m asl, is prone to single-stroke flashes. The origin of this discrepancy, compared to the other regions, is indeed worth a thorough investigation, but out of the scope of this particular study. Finally, Fleenor et al. (2009) found that 40% of the negative cloud-to-ground

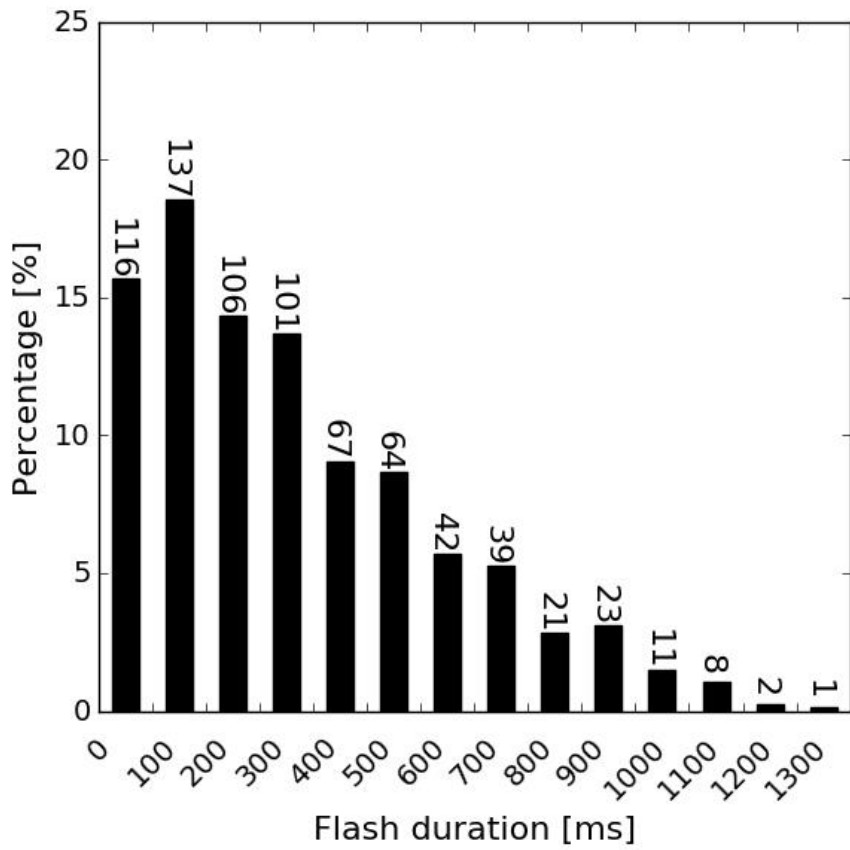

**Figure 3: Distribution of the flash duration in bins of 100 ms. The actual number of flashes within each bin is listed above the bars.**

175

flashes are single-stroke flashes observed in the U.S.A. Central Great Plains. It was noted that the time-resolution of the camera was limited to 16.7 ms, which could lead to an underestimation of the true negative multiplicity by about 11% (Biagi et al., 2007). However, even taken this underestimation into account, the percentage of the single-stroke flashes in Fleenor et al. (2009) is higher than the value in this study for US. The value of the maximum multiplicity per data set is indicated in Table

180 1 as well. One flash in SA stands out, containing a total of 26 strokes, while lasting 1.06 s.

As mentioned earlier, video observations allow classification of each stroke as a discharge creating either a new ground strike point (GSP), or following a PEC. As such, a total of 1832 GSPs are resolved within the different data sets; yielding an average of 1.56 GSPs per flash, while the mean amount of GSPs per flash for the different data sets ranges from 1.3 (SA) to 1.9 (BR). It follows that the average number of lightning strike points is 56% higher than the number of flashes. This value is in line

185 with those reported in earlier studies such as the 1.45 strike points per CG flash observed in Tucson, Arizona, by Valine and Krider (2002), 1.67 strike points per flash in Florida (Rakov et al., 1994), while in São Paulo, Brazil and in Arizona, U.S.A. a value of 1.70 was retrieved (Saraiva et al., 2010). The distribution of the number of GSPs per flash for the different data sets

is plotted in Fig. 2. SA is the data set containing the most amount of flashes with a single GSP percentage wise. This is a consequence of the amount of single stroke flashes observed in SA. In total, about 62% of the flashes strike ground at only one point. However, this value drops to 44% when single stroke flashes are excluded. In other words, the majority (56%) of multiple stroke negative downward flashes strike ground in more than one place. The maximum number of GSPs in a flash is found to be 5, observed in Austria as well as in South Africa. Finally, adopting the values in Table 1 for the multiplicity and average number of strike points for each data set, the average number of strokes observed per GSP varies between 1.82 (AT) and 2.94 (SA). For all the data sets combined it turns out that a ground contact point is struck 2.35 times on average.

Forked strokes, i.e., strokes whereby the lightning channel towards ground branches off, are an additional source of ground contact points. The occurrence of such strokes within each data set is indicated in Table 1. Averaged over all the data sets, it is found that 8.3% of the observed flashes comprise of at least one forked stroke. Examining those latter flashes that contain one or more forked strokes, 24.1% of the strokes within those flashes are forked, whereas overall this is only the case in 2.5% of all observed strokes in this study. If one would apply a percentage associated to the individual data sets of the observed strokes being forked, this results in an increase of the average amount of ground strike points per flash, N(GSP/flash), as indicated in Table 1, by this same factor.

Since the duration of a flash is defined as the time span between the first and last stroke in the flash, increased by the duration of an eventual continuing current following the last stroke, it is worthwhile to further highlight the occurrence and specifics of CCs. Following the approach as in Ballarotti et al. (2012), a 3 ms minimum CC duration is applied in order to eliminate what could just be return-stroke pulse tails in the high-speed camera records. Considering all ranges of CCs ($\geq$ 3 ms), the mean CC duration ranges from 38.5 ms in SA up to 67.1 ms as observed in AT, with an overall average of 44.5 ms. Median values are considerably lower with an overall median of 10 ms. The maximum value of 929 ms was measured in South Africa, which is about 200 ms longer than the maximum value found in Ballarotti et al. (2012). Out of 1096 flashes recorded with CC information, 51% contained continuing currents with duration greater than 10 ms and 57.7% of all strokes were followed by any CC greater than 3 ms. Only a small portion, i.e., 0.33%, of the strokes are followed by a CC longer than 500 ms.

Figure 3 illustrates the duration of all the flashes in bins of 100 ms. Since the US data set does not contain information on the possible occurrence of CC, the plot is made excluding US flashes. In addition, only multiple stroke flashes are included since many of the single stroke flashes were not followed by any CC, therefore influencing the percentage of flashes that fall in the first duration bin. The mean and median duration of multiple stroke flashes is found to be 371 ms and 313 ms, respectively. Ninety-five percent of the flashes have a duration below 926 ms. The flash with the longest duration of 1379 ms is observed in SA for a six stroke flash and is in line with the maximum flash duration values found in Saba et al. (2006) and Ballarotti et al. (2012).

One can intuitively suppose that with increasing flash multiplicity, the flash duration increases accordingly. While this is in general true, a large spread is observed in the data. This becomes apparent in Figure 4, which plots the flash duration as a

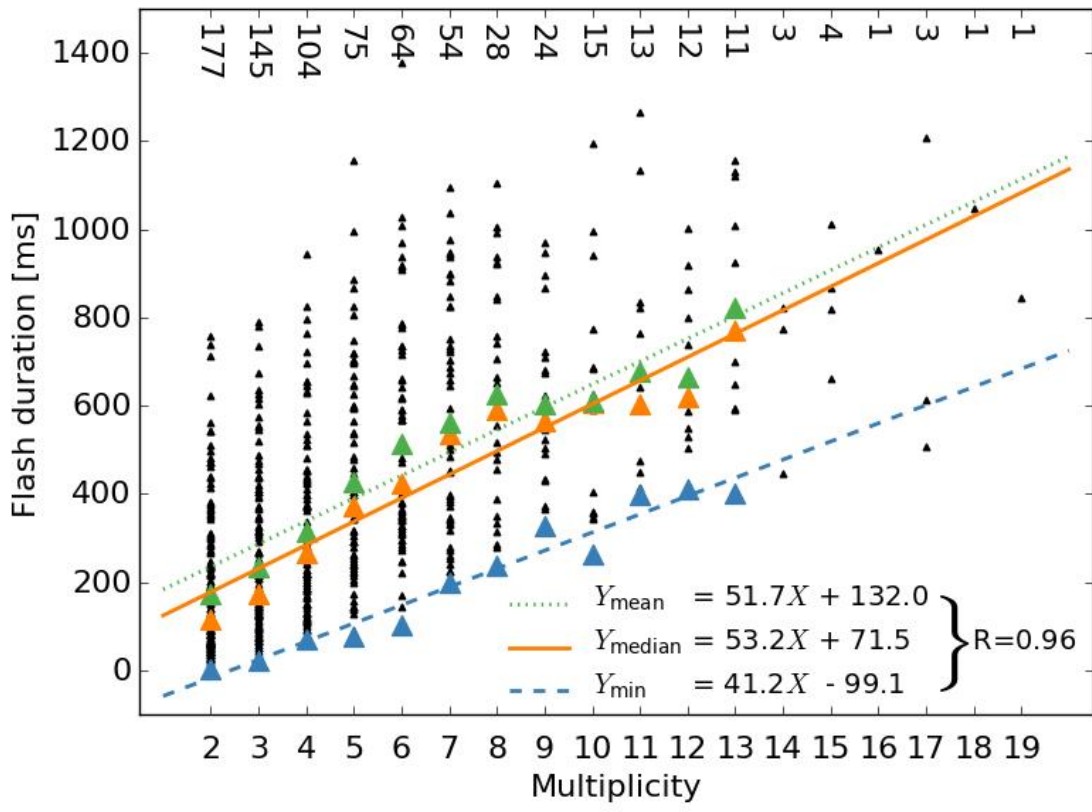

**Figure 4: Distribution of the flash duration as a function of multiplicity. The equation for the minimum, median and mean regression is given as well as the correlation coefficient of 0.96, being similar for all three regressions. The actual number of flashes per multiplicity is indicated at the top of the plot.**


function of multiplicity. Note that for instance in SA the maximum flash duration is found for a flash with multiplicity six. Additionally, Figure 4 indicates the regression slope based on the minimum, median and mean flash duration values per multiplicity. For this purpose, only multiplicities up to a value of 13 are taken into account since the sample size becomes too low at higher multiplicities. The regression equations, as well as the correlation coefficient, R, are indicated in the Figure. The

equations for the minimum and mean flash duration in this study compared to those presented in Saraiva et al. (2010) and Ballarotti et al. (2012) have a lower slope by a factor of 1.5 and 1.2, respectively.

Figure 5 displays the percentage of subsequent strokes creating a new GSP as a function of stroke order, based on a total of 658 new GSPs from the combined data sets. While a stroke with stroke order 2, i.e., the first subsequent stroke in the flash, still creates a new GSP in 60% of the cases, this quickly drops to 25% and 10% for the third and fourth stroke in the flash,

respectively. Those values are comparable to the values presented in Stall et al. (2009). On the other hand, although following a similar decreasing trend, the average percentage found in this study for a stroke with stroke order 2 in the flash is

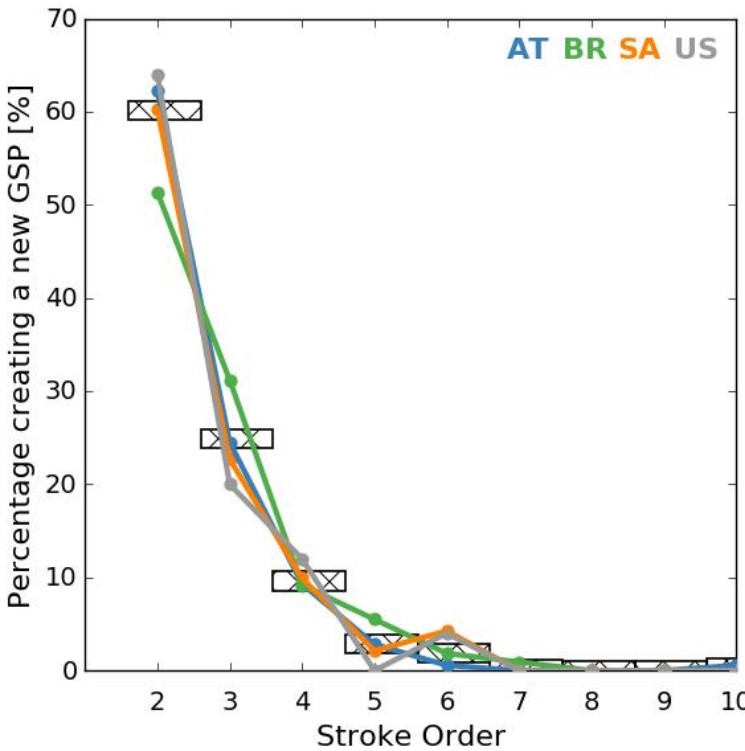

**Figure 5: Distribution of the percentage of subsequent strokes creating a new GSP as a function of stroke order. The shaded rectangles represent the result for the combined data sets.**

higher by about 10%-20% compared to what has been found previously in Rakov et al. (1994), Saba et al. (2006) and Ferro et al. (2012).

It has been suggested by Rakov and Uman (1990) that the conditions after the first stroke in the flash are not favourable to fully support the propagation of subsequent leaders all the way to the ground along the same path. Therefore, the stroke order alone is not sufficient enough to predict the chance of creating a new GSP, as the full lightning channels' history needs to be taken into account. The possibility to form a new lightning channel to ground as a function of the number of strokes that conditioned the previous lightning channel is quantified in Figure 6. Out of a total of 658 new lightning channels, 88.2%

developed after the occurrence of only one stroke, while this drops quickly to 7.6% and 2.6% in case of two and three observed consecutive strokes in the previous lightning channel, respectively. Note that in Austria two flashes are observed whereby a new GSP is created by the tenth stroke in the flash, while the lightning channel belonging to the previous GSP was used four and seven times, respectively. In the latter case, this indicates that even after seven consecutive strokes within the same lightning channel, it is still possible that the conditions to establish an unalterable path to ground are not met or are simply

ignored by a subsequent stroke. According to Ferro et al. (2012), when two or more strokes have used the previous lightning

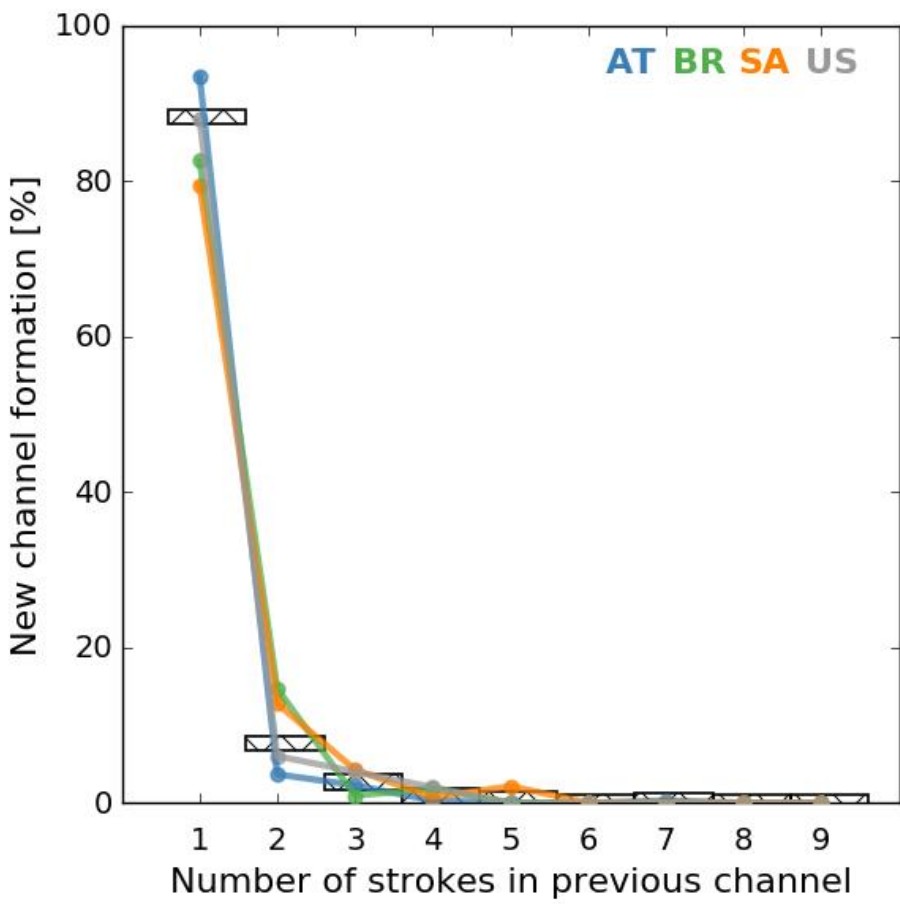

**Figure 6: Relation between new lightning channel formation and the number of strokes in previous lightning channel. The shaded rectangles represent the result for the combined data sets.**

channel, then a larger interstroke time interval may be an important factor in the creation of a new lightning channel. While the interstroke time intervals will be discussed in more detail later on, it is worthwhile to point out that the interstroke time intervals between the ninth and tenth stroke in case of the two Austrian flashes as mentioned above are 26.2 ms and 103.97 ms, respectively.


The distribution of 3128 time intervals is plotted in Figure 7 adopting a bin size of 20 ms and results thereof are listed in Table 2. The average time interval is 85 ms, with a geometric mean (GM) of 57 ms. The maximum time interval for the individual data sets is in the order of 500 ms to 700 ms, except for SA which contains a six-stroke flash with a maximum observed time interval of 905 ms between the fifth and last stroke in the flash. Note that this particular flash is also the flash with the maximum flash duration in all the data sets, and can be regarded as an exception, although time intervals well exceeding 500 ms are


recorded in other studies, e.g., Saba et al. (2006). Usually, these long time intervals between strokes are due to a very long continuing current event following the first one. The 99[th] percentile appears to be 470 ms, somewhat below the standard

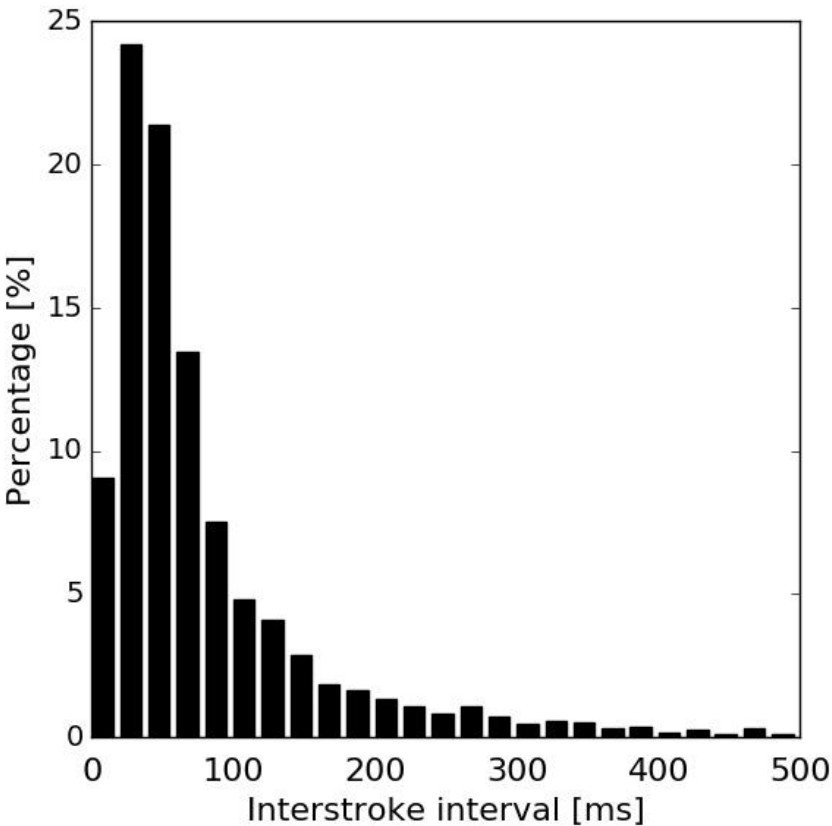

**Figure 7: Interstroke time intervals for all subsequent strokes and for same and new lightning channels.**

maximum interstroke time criterion of 500 ms usually adopted to group different strokes into flashes by lightning location
systems.

It is possible to further separate the interstroke time intervals from Figure 7 into intervals preceding strokes down the same
lightning channel, $\Delta T_{PEC}$, or down a new lightning channel, $\Delta T_{NGC}$. The results of this can be viewed in Table 2 for the
individual data sets, as well as for all the data sets combined. Overall, it is found that the GM for $\Delta T_{NGC}$ is slightly larger
compared to $\Delta T_{PEC}$ by 10 ms. While Rakov et al. (1994) found a larger difference between $\Delta T_{NGC}$ and $\Delta T_{PEC}$, this was probably
due to the limited sample size involved. Subsequent follow-up studies by, e.g., Saba et al. (2006) and Ferro et al. (2012),
showed that the GM values of $\Delta T_{NGC}$ and $\Delta T_{PEC}$ are converging towards each other while adopting a larger data set, as is the
case in this study.

**Table 2. Statistics for interstroke time intervals that precede subsequent PEC and NGC**

| | AT | | BR | | SA | | US | | ALL | | |
|---|---|---|---|---|---|---|---|---|---|---|---|
| | N* | GM, ms | N | GM, ms | N | GM, ms | N | GM, ms | N | GM, ms | SE**, ms |
| $\Delta T_{PEC}$ | 662 | 62 | 362 | 68 | 1199 | 49 | 162 | 52 | 2385 | 55 | 1.8 |
| $\Delta T_{NGC}$ | 351 | 56 | 108 | 64 | 133 | 93 | 42 | 73 | 634 | 65 | 3.9 |
| $\Delta T_{All}$ | 1013 | 60 | 470 | 67 | 1332 | 52 | 204 | 57 | 3019 | 57 | 1.6 |

*N= sample size

**SE = Standard Error

There are some noticable differences among the individual data sets. While it is clear that $\Delta T_{NGC}$ is considerably larger than $\Delta T_{PEC}$ in SA and US, the differences are much smaller or the opposite in the other data sets.

Some further investigation with respect to the time differences, analogous to Ferro et al. (2012), are presented in the following. From Fig. 6 it is found that in 88% of the cases a new lightning channel formation is observed after just one stroke in the previous lightning channel. Investigating now the time intervals in the other 12% of the cases when two or more strokes re-

used the previous lightning channel, we find that the average interstroke time interval preceding a new lightning channel becomes 77 ms, compared to a time difference of 34 ms between strokes that follow the same lightning channel, see Table 3. Therefore, in this subset of flashes, $\Delta T_{NGC}$ is about 2.3 times larger compared to $\Delta T_{PEC}$. This value is somewhat lower compared to the 3.5 times found in Ferro et al. (2012), but still of the same order. Note that the interstroke time interval GM value for PEC strokes is in this case lower by a factor of 1.6 compared to the result in Table 2.


**Table 3. Interstroke time interval between strokes using a PEC and interstroke time interval preceding a NGC after two or more strokes down the same lightning channel**

| | AT | | BR | | SA | | US | | combined | |
|---|---|---|---|---|---|---|---|---|---|---|
| | N* | [ms] | N | [ms] | N | [ms] | N | [ms] | N | [ms] |
| PEC | 38 | 31 | 24 | 44 | 56 | 32 | 10 | 31 | 128 | 34 |
| SE**[ms] | 8.8 | | 6.2 | | 5.5 | | 6.4 | | 3.8 | |
| NGC | 23 | 68 | 19 | 67 | 29 | 86 | 6 | 113 | 77 | 77 |
| SE [ms] | 15.3 | | 15.6 | | 33.7 | | 19.2 | | 14.4 | |

*N = sample size

**SE = Standard Error

## 4. Summary

Ground strike point characteristics in negative ground lightning flashes have been investigated by means of high-speed camera observations taken in different parts around the globe. According to our knowledge this is the first simultaneous analysis of GSP properties in different regions of the world applying a common methodology. It is found that the mean amount of ground strike points per flash is 1.56, varying in the four regions from 1.29 to 1.90. The maximum number of GSPs per flash just fluctuates between 4 and 5, while the mean number of strokes per GSP ranges from 1.82 to 2.94. From this, it follows that the ground strike point statistics differ in different regions. The values quoted in this study are in line with those found in the literature, and reconfirms the necessity to take ground strike points into account to estimate the risk for lightning protection purposes. While the number of flashes and strokes involved in this study is statistically relevant and, above all, larger compared to any other similar study undertaken in the past, it remains a snapshot of that particular moment in time and place. Consequently, it requires investigation in more detail of the regional and seasonal trends that might exist. In order to overcome this, one could make use of the observations made by LLSs. Present day LLSs provide, with a high degree of accuracy both in terms of efficiency and location, the different strokes that compose a flash. Ingesting those observations into a so called ground strike point algorithm, in order to group individual strokes into ground strike points, would provide a means to study on a larger temporal and spatial scale the characteristics of ground strike point densities. The interested reader is referred to Poelman et al. (2021, nhess-2021-13 companion paper) to learn more about the ability of three such algorithms to determine the observed ground strike points correctly based on the data set presented in this study.

The 99[th] percentile of the interstroke intervals is found to be 470 ms and certifies the commonly used maximum interstroke interval of 500 ms to group strokes observed by a LLS into a flash while adopting a certain distance threshold. In addition, it follows that the GM value for time intervals preceding the occurrence of a new lightning channel is only slighlty larger than the typical GM interstroke interval value of 57 ms. Overall, apart from a few exceptions, the total flash duration is below one second and exhibits a positive correlation with the flash multiplicity.

In the majority of the cases, i.e., 88%, a new lightning channel formation is observed after just one stroke in the previous lightning channel. This fact, together with the almost similar interstroke time intervals preceding strokes producing a NGC or following a PEC, suggests that time interval alone is not enough to influence the creation of a new lightning channel to ground. However, examining the cases when two or more strokes re-used the previous lightning channel, the average interstroke time interval preceding a new lightning channel is more than double the interval time between previous strokes that follow the same lightning channel. This analysis strengthens the outcome of Ferro et al. (2012).

## Data availability

All data processed could not be available for public. For the access, the first author can be contacted by email: dieter.poelman@meteo.be

## Author Contributions

DRP and WS conceptualized the research. DRP and WS carried out the analysis with contributions from SP, DH, MS and HH. WS, DH, MS, HH, LS, CV, CTM, CS and TW are strongly involved in the collection and preparation of the used data sets. DRP prepared the manuscript with review and editing from all co-authors.

## Competing interests

The authors declare that they have no conflict of interest.

## Acknowledgments

The work done by the reviewers of this article proved invaluable. Their critical reading and contribution of ideas and comments are very much appreciated by the authors. We would like to thank them sincerely for their time and effort. HH and CS would like to thank the National Research Foundation of South Africa (Unique Grant No: 98244).

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
