# Peer review of "Global ground strike point characteristics in negative downward lightning flashes — part 1: Observations"

_Natural Hazards and Earth System Sciences, 2021_

## Author Comment (AC1)

**General comments:**

This paper presents a comprehensive study of negative downward lightning flashes based on high-speed video camera recordings of negative cloud-to-ground lightning in several regions around the globe. This study presents solid statistics that help improve the current lightning protection standard (change from flash density to ground-strike point density). The subject is suitable for this journal. Several comments follow. I recommend this paper be accepted after minor revisions.

**Specific Comments:**

1. Did the authors include upward lightning in South Africa dataset? If yes, I think those upward lightning contradicts your title (negative downward flashes). If not, please state so in the paper.

   **=> Upward lightning flashes are not taken into account in the South African data set. This will be highlighted in the next version of the manuscript.**

2. "Note that in Austria two flashes are observed whereby a new GSP is created by the tenth stroke in the flash, while the channel belonging to the previous GSP was used four and seven times, respectively." It would be interesting to know the interstroke interval preceding the $10^{th}$

   **=> The interstroke interval preceding the $10^{th}$ stroke in the two flashes are 26.2 ms and 103.97 ms, respectively. This will be indicated in the next version of the manuscript.**

3. Flash characteristic studies solely relying on high-speed cameras have limitations. I hope the authors could discuss those limitations and how those limitations could possibly influence the statistics presented. Two limitations that I can think of: (1) strokes creating a new termination could be missed by the camera (e.g., the stroke can occur at the back of cameras or simply out of view). (2) It is likely camera record length is not long enough to cover the entire flash. I see that length for SA dataset is only 1s with manual trigger setup (not sure what's the pre-trigger and post-trigger during manual trigger setup), maybe this partially explains why most SA flashes are single-stroke flash. Simultaneous electric/magnetic field measurements/LLS data might help mitigate some of those limitations. They could be used to see if there are additional strokes in the vicinity but outside the field of view of camera or outside the duration of the camera records.

   **=> The introductory paragraph of Sect. 2 'Data acquisition and analysis' will now include the limitations linked to high-speed camera observations of lightning as suggested by the reviewer:**

   **"Ground-truth campaigns are time consuming in order to gather enough data to be statistically relevant. To reach this objective, ground-truth datasets are collected from**

**different geographical regions and taken over various periods in time: Austria (AT) in 2012, 2015, 2017 and 2018, Brazil (BR) in 2008, South Africa (SA) in 2017-2019 and U.S.A. (US) in 2015.**

**It is of importance to recognize the limitations inherent to high-speed camera observations when used in flash characteristic studies. In particular, strokes creating a new termination could be missed by the camera when occurring out of the camera's field of view. In addition, the record length should be long enough in order to capture the entire flash, i.e., typically longer than one second. Aiming to minimize as much as possible the influence of the latter on flash statistics, high-speed camera observations should be checked against concurrent electric field measurements to ensure a stroke was not missed. In this, flashes with channels that are outside the field of view can be excluded from the data. For the measurements in all of the data sets presented in this study electric field measurements have been used, and therefore only flashes, where a clear visible channel to the ground is observed for all the associated strokes are included. However, it should be noted that even though such a selection of flashes is made, it does not undeniably resolve the true contact point all of the time. This is certainly true when the observations are made at ground level or even worse in the Alps. As such, the amount of ground strike points retrieved from the video fields as discussed later on in this study should be regarded as a lower limit. In the cases where the time interval between subsequent strokes is lower than 1 ms, the stroke is considered to be a forked stroke rather than a stroke creating a new GSP, which in turn reduces the multiplicity of the flash. All the data sets, except US, indicate the duration of the continuing current (CC) for each stroke if present in the recorded video fields.**

**In what follows, a description is given of the instrumentation set-up used at the different regions and the periods of investigation.***"*

**=> Related to SA: The buffer time is not one second (as written in the original manuscript), but 1.8 second. We will adapt Sect. 2.3 as follows: "… The setup utilizes two high-speed cameras (a Phantom v7.1 and a Phantom v310) which are located North-West of the city. Frame rates used are in the range of 5000 to 15000 fps and all captured videos are GPS time-stamped. A 1.8 second buffer time is used and events are manually triggered. Typically, the pre-trigger and post-trigger were set approximately 60/40 of the 1.8 second buffer respectively. …."**

4. "It follows that the channel creating a GSP is re-used by a factor of 2.3" I think the word "re-used" is ambiguous. Sounds like the termination created by a previous stroke will be re-struck by 2.3 subsequent return strokes on average. Your statement "A ground contact point is struck 2.35 times on average" in Line 166 is more accurate.

**=> This particular sentence in the abstract will be simply rewritten as 'It follows that a ground contact point is struck 2.35 times on average.'**

**Minor editorial suggestions:**

1. Line 65: "Hence, the role of high-speed camera observations." This Is not a complete sentence.
2. Line 70, enable us to determine
3. L127, 150 m

   => **Editorial suggestions will be taken into account**

---

## Author Comment (AC2)

**General comments:**

The authors provide a nice, concise summary of high-speed video observations of negative CG flashes from four locations around the world. My only pre-publication suggestions would be perhaps to consider augmenting the paper with some additional information that apparently should be in the data sets. This includes:

- lines 90-92: statistics on the occurrence of forked strokes, and on the occurrence of and duration of continuing currents (at locations other than the US site, as noted)

**=> Investigating the characteristics of forked strokes is indeed interesting. Unfortunately, the indication of forked strokes was only properly documented for the Austrian 2018 data set, and (almost) not for the others. It would require a considerable effort to add information about forked strokes for all the data sets. Hence, we believe this is more appropriate for a potential follow-up study.**

**Anyhow, based on AT 2018 it is found that 9.4% of the observed flashes comprise of at least one forked stroke (18/191). This is in line with the 12.4% found in Valine and Krider (2002)\* whereby 48 (37 + 11) out of a total of 386 cloud-to-ground lightning flashes contained forked strokes or produced channels "that struck the ground in two or more places and shared a common channel at higher altitudes". Examining those latter flashes that contain forked stroke(s) in AT 2018, 34.4% of the strokes within those flashes are forked (21/61), whereas overall this is only the case in 3.75% of all observed strokes (21/560). If one would apply a similar amount of ~3-4% of the observed strokes being forked in the other data sets, this results in an increase of the average amount of ground strike points per flash, N(GSP/flash), as indicated in Table 1 by this same factor.**

**This will be included in the new version of the manuscript.**

**=> Investigating the presence of continuing current (CC) leads to following findings:**

| | Continuing Current (CC) | | | | |
|---|---|---|---|---|---|
| | AT | BR | SA | US | ALL |
| Mean (ms) | 67.1 | 41.2 | 38.5 | / | 45.8 |
| Median (ms) | 15.0 | 10.0 | 9.0 | / | 10.0 |
| Max (ms) | 540 | 540 | 929 | / | 929 |
| % strokes followed by CC ≥3ms | 33.7 | 48.1 | 73.0 | / | 54.0 |
| % strokes followed by CC >500ms | 0.26 | 0.49 | 0.38 | / | 0.35 |
| % flashes containing CC >10ms | 37.8 | 52.8 | 61.8 | / | 50.1 |

Following the approach as in Ballarotti et al. (2012)*, a 3 ms minimum CC duration is applied in order to eliminate what could just be return-stroke pulse tails in the high-speed camera records. Considering all ranges of CC ($\geq$ 3 ms), the mean CC duration ranges from 38.5 ms in SA up to 67.1 ms as observed in AT, with an overall average of 45.8 ms. Median values are considerably lower with an overall median of 10 ms.  The maximum value of 929 ms was measured in South Africa, which is about 200 ms longer than the maximum value found in Ballarotti et al. (2012).

Out of 1096 flashes recorded with CC information, 50.1% contained continuing currents with duration greater than 10 ms and 54% of all strokes were followed by any CC greater than 3 ms. Only a small portion, i.e., 0.35%, of the strokes are followed by a CC longer than 500 ms.

* Ballarotti, M. G., C. Mediros, M. M. F. Saba, W. Schulz, and O. Pinto Jr. (2012), Frequency distributions of some parameters of negative downward lightning flashes based on accurate-stroke-count studies, J. Geophys. Res., 117, D06112, doi:10.1029/2011JD017135.

This will be included in the new version of the manuscript.

- lines 160-161 and other places where the data from South Africa are discussed: comparisons of upward-initiated flashes from the Sentech and Hillbrow towers in Johannesburg with observations from other towers, such as Gaisberg, Peissenberg, and Santis in the Alps, or the CN tower in Toronto. The comparison with the towers in the Alps might be particularly interesting insofar as so much of their lightning occurs in winter, if I am not mistaken, whereas the Johannesburg site is noted in this manuscript as having essentially no winter lightning.

=> Upward lightning flashes are not taken into account in the South African data set. This will be highlighted in the next version of the manuscript. Therefore, the comparison with upward-initiate flashes in other locations, e.g., Alps or Toronto, is out of the scope of this work.

- also related to lines 160-161: is the fact that the South African data set has a much larger percentage of single-stroke flashes than the others partly influenced by the population of upward-initiated flashes in that data set? It would be interesting to know, one way or the other

=> As noted above, all upward flashes in SA are extracted from the SA data set in this work.

=> The percentage of single-stroke flashes as described in the 2013 CIGRE Brochure 549 (Table 2.1) range from 13% (New Mexico; Kitagawa et al., 1962) up to 21% (Sri Lanka; Cooray and Jayaratne, 1994). The values found in this manuscript (Table 1) are somewhat higher ranging from 23% (BR) up to 38.4% (SA).

Note that in our study flashes are removed if at least one channel is partly visible, diffuse or simply out of the field of view. As such only flashes are kept of which the different GSPs in the video images could be determined with great confidence. The number of flashes that are removed from the data sets is anyhow minimal w.r.t. total number of flashes per data set. However, taking those removed flashes into account to re-calculate the percentage of single-stroke flashes it turns out that the percentage in case of AT drops from 29.2% to 28.2%, for BT from 23% to 21.4%, for SA from 38.4% to 37.9%, whereas it remains unchanged for US at 25.6%.

For AT: We would like to draw your attention to the recently published article by Schwalt et al. (2021)* in which the authors investigate specifically the percentage of single-stroke flashes in Austria. It is found that the percentage of single-stroke flashes among all negative flashes is 27%. A possible dependency of the occurrence of single-stroke flashes with the underlying terrain (Alpine versus pre-Alpine) is found in this study. The 28.2% found in the present study is therefore in line with the findings of Schwalt et al. (2021).

For BR: the newly calculated percentage of 21.4% is only slightly higher compared to the 17% quoted in the 2013 CIGRE brochure 549.

For SA: Looking into the LLS data (thus not just the correlated high-speed camera cases) in a corresponding area as in this study and averaged over a few years, a similar value of 1.2 strokes per flash is found. It seems that this area, at an altitude of about 1600 m asl, is prone to single-stroke flashes. The origin of this discrepancy, compared to the other regions, is indeed worth a thorough investigation but out of the scope of this study.

For US: We would like to draw your attention to Fleenor et al. (2008)**. In this study 40% (41/103) of the negative cloud-to-ground flashes are single-stroke flashes. It was noted that the time-resolution of the camera was limited to 16.7 ms, which could lead to an underestimation of the true negative multiplicity by about 11% (Biagi et al., 2007). However, even taken this underestimation into account, the percentage of the single-stroke flashes in the present study is still in line with Fleenor et al. (2008).

*Schwalt, L., Pack, S., Schulz, W., and Pistotnik, G. (2021). Percentage of single-stroke flashes related to different thunderstorm types, Electric Power System Research, 194, 107109

**Fleenor, S. A., Biagi, C. J., Cummins, K. L., and Krider, E. P. (2008). Characteristics of cloud-to-ground lightning in warm season thunderstorms in the Great Plains, 20th International Lightning Detection Conference, 21-23 April, Tucson, Arizona, USA.

- The only other comment is about references in lines 48-49 – at least some are outdated, and should be updated where possible; the Cummins et al. reference on the NLDN is especially old, and in fact, there's a brand new paper in the March 2021 Journal of Atmospheric and Oceanic Tech. about the 2013++ NLDN.

=> Cummins et al. (1998) will be replaced by 'Murphy, M. J., Cramer, J. A., and Said, R. K.: Recent History of upgrades to the U.S. National Lightning Detection Network, Journal of Atmospheric and Oceanic Technology, https://doi.org/10.1175/JTECH-D-19-0215.1'

=> Lay et al. (2004) will be replaced by 'Burgesseur, Rodrigo E., Assessment of the World Wide Lightning Location Network (WWLLN) detection efficiency by comparison to the Lightning Imaging Sensor (LIS), Q. J. R. Meteorol. Soc. 143: 2809–2817, October 2017 A DOI:10.1002/qj.3129'

=> Thomas et al. (2004) will be replaced by 'Coquillat, S., Defer, E., de Guibert, P., Lambert, D., Pinty, J.-P., Pont, V., Prieur, S., Thomas, R. J., Krehbiel, P. R. and Rison, W., SAETTA: high-resolution 3-D mapping of the total lightning activity in the Mediterranean Basin over Corsica, with a focus on a mesoscale convective system event, Atmospheric Measurement Techniques, 12, 2019, doi:10.5194/amt-12-5765-2019.'